# ArborKV: Structure-Aware KV Cache Management for Scaling Tree-based LLM Reasoning

**Yeqiu Chen** [* 1 2]  **Ziyan Liu** [* 1]  **Zhenxin Huang** [1]  **Runquan Gui** [1]  **Hong Wang** [1]  **Lei Liu** [1 †]

## Abstract

Recent progress in LLM reasoning has increasingly shifted from single-pass generation to explicit search over intermediate reasoning states. Tree-of-Thoughts (ToT) organizes inference to tree-structured search with branching and backtracking, but it substantially amplifies the Key–Value (KV) cache: retaining KV states for a frontier of partial trajectories quickly becomes a memory bottleneck that limits throughput and constrains search depth and width under fixed hardware budgets. We address this challenge by observing that KV reuse in ToT-style inference is governed by search dynamics: near-term decoding depends primarily on the active branch and its ancestors, whereas inactive subtrees have low short-term reuse probability yet must remain recoverable for backtracking. Motivated by this, we propose **ArborKV**[1], a structure-aware eviction framework that couples a lightweight value estimator with a tree-aware allocation policy, and performs purely token-extractive eviction with lazy rehydration to support revisits. Experiments on ToT-style reasoning benchmarks show that ArborKV achieves up to $\sim 4\times$ peak KV-memory reduction while preserving near-full-retention accuracy, enabling larger search configurations under fixed device budgets that would otherwise run out of memory.

## 1. Introduction

Recent progress in Large Language Model (LLM) reasoning has increasingly shifted from single-pass generation to

---

[*]Equal contribution , [§]Work done during an internship at Huawei. [1]University of Science and Technology of China [2]Huawei Technologies Co., Ltd.. Correspondence to: Lei Liu <liulei13@ustc.edu.cn>.

*Proceedings of the $43^{rd}$ International Conference on Machine Learning*, Seoul, South Korea. PMLR 306, 2026. Copyright 2026 by the author(s).

[1]"Arbor" is the Latin word for "tree."

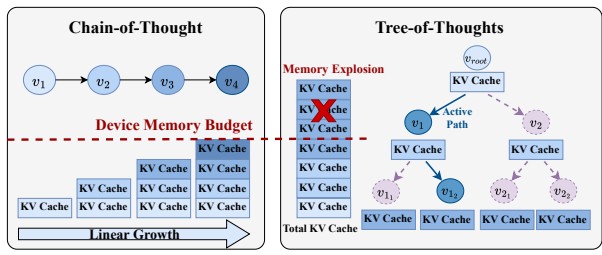

**(a)** Memory Pressure: Linear vs. Tree Search

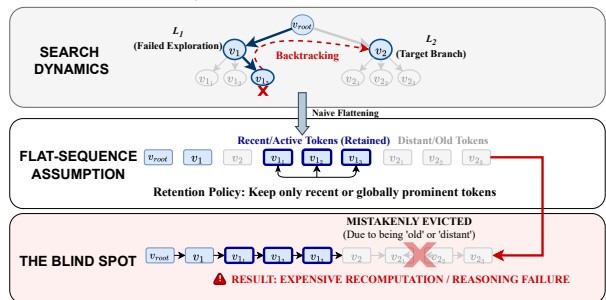

**(b)** Prior Art: Naive Flattening Blind Spot

**Figure 1. Challenges in Tree-structured Reasoning.** (a) Transitioning from linear CoT to tree-based search leads to a KV-cache memory explosion that exhausts hardware budgets. (b) Sequence-centric policies flatten the tree into a linear pool, mistakenly evicting inactive branches required for future backtracking.

*explicit search* over intermediate reasoning states. Chain-of-Thought (CoT) prompting elicits a linear sequence of rationales to improve multi-step reasoning (Wei et al., 2022), and Tree-of-Thoughts (ToT) formalizes inference as searching over multiple candidate thought states, explicitly modeling branching and backtracking (Yao et al., 2023). This paradigm has been widely adopted in practice, with modern inference engines further scaling ToT via parallel tree-search controllers that maintain a frontier of candidates and dynamically transition the search focus (e.g., DPTS) (Ding et al., 2025). However, such tree-based inference introduces a prohibitive system bottleneck: the transformer key–value (KV) cache must sustain autoregressive decoding for numerous partially expanded paths, where the peak KV footprint can quickly exhaust available device memory (see Figure 1a).

To facilitate efficient decoding, decoder-only transformers cache per-layer attention keys and values for previously pro-

cessed tokens. As a result, KV memory consumption grows linearly with the sequence length and scales with model depth and attention dimensions (Shazeer, 2019; Kwon et al., 2023). Tree-based reasoning exacerbates this cost beyond standard linear decoding: ToT-style search necessitates retaining the KV states for an entire frontier of partial trajectories to enable seamless backtracking and exploration of alternative branches (Yao et al., 2023). Parallel controllers such as DPTS magnify this peak-memory pressure as multiple candidates are activated concurrently (Ding et al., 2025). Consequently, naively materializing KV states for all nodes in a reasoning tree can quickly exhaust GPU memory, forcing practitioners to trade off between reducing search budgets (depth/width), shrinking batch sizes, or applying aggressive compression—often at the expense of throughput or reasoning performance.

Although a substantial body of work has explored inference-time KV management, these methods remain largely misaligned with the non-linear dynamics of thought-tree reasoning. System-level approaches like PagedAttention (Kwon et al., 2023) optimize KV storage to mitigate fragmentation, yet remain agnostic to the semantic importance of content under tight budgets. Algorithmic cache policies typically focus on token-level salience within a flat sequence—identifying "heavy hitters" (Zhang et al., 2023), attention sinks (Xiao et al., 2023b), or using prompt-guided cues (Corallo & Papotti, 2024) to prune less-contributive tokens. While more recent work like ThinKV (Ramachandran et al., 2025) shifts focus toward "thought-aware" compression by exploiting attention sparsity specific to reasoning steps, it is fundamentally designed for single linear trajectories. Consequently, these methods either (i) optimize throughput while leaving allocation semantics untouched (Kwon et al., 2023; Aminabadi et al., 2022; Sheng et al., 2023; Patel et al., 2024), or (ii) assume a monolithic, forward-moving path, failing to model the topological constraints and frequent backtracking inherent in tree-structured search (Yao et al., 2023; Zhou et al., 2023; Ding et al., 2025), as illustrated in Figure 1b.

We address these limitations by recognizing that KV-cache reuse in tree-based inference is fundamentally governed by search dynamics rather than token-level statistics. Concretely, the cacheable unit is a *thought block*—contiguous spans representing discrete reasoning steps, and autoregressive expansion depends exclusively on the ancestor chain of the active leaf, while inactive branches remain latent for potential backtracking. Unlike conventional sequence-centric policies that fail to balance the competing demands of branch exploration and backtracking, our approach introduces a *tree-aware allocation principle*. Specifically, KV retention should dynamically track the shifting search focus, prioritizing blocks based on both (i) their semantic utility to the reasoning process and (ii) their topological proximity to the active search frontier, all while maintaining the low-latency requirements of online inference.

Building on this tree-aware principle, we propose **ArborKV**, a *structure-aware eviction framework* comprising two synergistic components: a *Multi-Signal Value Estimator* (MSVE) and a *Tree-Aware Eviction* (TAE) policy. Specifically, MSVE quantifies the utility of each thought block by fusing the search engine's strategic feedback with the model's internal generation confidence, identifying essential reasoning steps without incurring additional computational overhead. TAE then maps these utility scores onto the tree topology, dynamically allocating retention budgets based on a block's proximity to the active search frontier. This framework is further supported by a *lazy rehydration* mechanism to efficiently restore evicted states during backtracking. Furthermore, we formalize this allocation strategy as a *constrained optimization problem*, grounding the policy's balance between semantic merit and geometric relevance under strict memory budgets.

In summary, our contributions are threefold.

(i) We formulate KV-cache management for structured reasoning as a **tree-aware optimization problem**. We propose a framework that couples a **TAE** policy with an intra-block token-extractive mechanism, explicitly aligning memory allocation with the branching and backtracking dynamics of tree-based search.

(ii) We introduce **MSVE**, a low-overhead scoring module that quantifies the utility of thought blocks by fusing search-level strategic signals, semantic uncertainty, and attention statistics. This enables calibrated importance estimation without requiring additional online forward passes.

(iii) Extensive experiments on ToT-style reasoning benchmarks (GSM8K, SVAMP, and Game of 24) demonstrate that ARBORKV delivers up to $\sim 4\times$ peak KV-cache reduction while maintaining near-full-retention reasoning accuracy, thereby enabling larger tree-search settings under fixed device budgets that would otherwise run out of memory.

## 2. Related Works

### 2.1. Tree-Structured Reasoning with Large Language Models

LLM reasoning has transitioned from monotonic decoding to explicit search (Yao et al., 2023; Gui et al., 2025). While Chain-of-Thought (CoT) (Wei et al., 2022) and self-consistency (Wang et al., 2022) enhance multi-step reasoning through linear trajectories, Tree-of-Thoughts (ToT) (Yao et al., 2023) and formalize search via branching and backtracking. Modern engines like DPTS (Ding et al., 2025) scale these via parallel controllers, yet the concurrent maintenance of multiple paths creates a prohibitive KV-cache bottleneck that constrains search budgets (Ding et al., 2025).

## 2.2. KV-Cache Management for Efficient Inference

**System and Architectural Optimizations.** Prior research on KV-cache optimization spans system-level infrastructure and architectural modifications. PagedAttention (Kwon et al., 2023) improves throughput by mitigating fragmentation via virtual memory principles, while FlexGen (Sheng et al., 2023) explores offloading and compression to run large models under strict GPU constraints. Architectural approaches, such as Multi-Query Attention (MQA) (Shazeer, 2019) and Grouped-Query Attention (GQA) (Ainslie et al., 2023), reduce the KV footprint through parameter sharing but typically require architectural commitments or retraining. Furthermore, quantization methods like LLM.int8() and SmoothQuant reduce cache memory at the cost of some precision (Dettmers et al., 2022; Xiao et al., 2023a). While kernel-level optimizations like FlashAttention (Dao et al., 2022) accelerate computation, they are orthogonal to the **retention policy**—the decision of which states to preserve under constrained memory.

**Algorithmic Eviction and Compression.** A significant body of work performs test-time eviction under a **sequence-centric assumption**. Representative approaches retain tokens that appear important according to attention or related proxies, including heavy-hitter retention (Zhang et al., 2023), attention-sink stabilization (Xiao et al., 2023b), persistence-based importance (Liu et al., 2023), and prompt-guided compression (Corallo & Papotti, 2024). More recent methods refine the granularity of decisions: Keyformer selects a subset of "key tokens" to reduce KV bandwidth/footprint (Adnan et al., 2024), adaptive KV cache construction profiles head behaviors to compress KV on the fly (Ge et al., 2023), and Ada-KV allocates *head-wise* budgets with a principled objective tied to eviction loss (Feng et al., 2024). While effective for linear generation, these methods fail to model the **topological constraints** and non-monotonic focus shifts inherent in tree search. Recent "thought-aware" approaches like ThinKV (Ramachandran et al., 2025) adapt compression to reasoning steps, but remain optimized for single linear trajectories. In contrast, ArborKV optimizes KV allocation directly over the tree structure, utilizing a reversible eviction mechanism that accommodates the non-monotonic nature of search-based inference.

## 3. Preliminaries

We model the decoding trajectory of a long-horizon inference as a rooted tree $\mathcal{T} = (V, E)$ of *thought blocks*. Each node $i \in V$ corresponds to a contiguous token span $x_{a_i:b_i}$ produced during one locally coherent step of reasoning; let $n_i = b_i - a_i + 1$ denote its length. The tree evolves online as the search procedure (e.g., ToT/DPTS-style exploration) expands one *active leaf* $\ell^\star$ at a time while preserving pre-viously explored branches for potential backtracking. For any node $i$, let $d_i$ be its depth (root at depth 0), and let $\Delta_i$ be the shortest-path distance in $\mathcal{T}$ from $i$ to $\ell^\star$; the ancestor chain from root to $\ell^\star$ is denoted $\mathrm{Path}^\star$.

The attention KV cache stores a per-token state whose memory cost is approximately constant across tokens at a fixed model configuration. We therefore express the instantaneous memory usage as

$$M \propto \sum_{i \in V} k_i, \qquad 0 \le k_i \le n_i,$$

where $k_i$ denotes retained tokens for block $i$. Under a global budget $M \le \mathcal{B}$, the policy dynamically determines $k_i$ and selects specific tokens to preserve. We enforce three invariants for stability:

**(i) Active-path protection.** For $i \in \mathrm{Path}^\star$, we require $k_i = n_i$ or a high floor $k_{\mathrm{protect}}$.

**(ii) Per-block minimum.** For all $i$, $k_i \ge K_{\min}$ to prevent catastrophic "thread drop," where $K_{\min}$ is a count-level lower bound on the number of retained tokens for each block. We also maintain a small tail window $L_{\mathrm{tail}}$ that preserves immediate local context, where $L_{\mathrm{tail}}$ denotes the number of most recent tokens that are always kept within each block.

**(iii) Head-only eviction.** Within a block, we keep the *most recent* tokens and evict earliest tokens first; this preserves the local Markovian support for the next decoding step.

We update the KV cache only at three *policy update events* (PUEs); at each PUE we recompute retention targets $k_i$ via Eqs. 2–3 and then apply token-*extractive* evictions. (i) *Thought boundary* (a block $i$ closes by end-of-span or explicit tag): finalize node $i$, form features $\phi_i$, obtain $s_i$, set $k_i$, and evict the earliest $n_i - k_i$ tokens *of that block only*. (ii) *Transition* (the active leaf switches when the search expands a new child or backtracks): recompute all tree distances $\Delta_j$, reallocate budgets with sibling coordination, *pin* the new ancestor chain $\mathrm{Path}^\star$, and evict on non-active sub-trees wherever the new $k_j$ decreases. (iii) *Memory pressure* ($M \ge \mathcal{B} - \delta$ for a small safety margin $\delta$): tighten global knobs (e.g., lower $\alpha$ or raise $\lambda_\Delta$), recompute $k_j$, and evict lowest-priority off-path blocks until $M \le \mathcal{B}$. *Rehydration*: if a previously evicted block becomes part of $\mathrm{Path}^\star$, rebuild its KV *before* the next token step by a single prefill over $x_{a_i:b_i}$ (no token generation), then resume decoding.

## 4. Method

We propose a structure-aware memory management framework (Figure 2) that explicitly models the inference process as a rooted tree $\mathcal{T} = (V, E)$ of thought blocks. ArborKV couples a *Multi-Signal Value Estimator* (MSVE) that scores

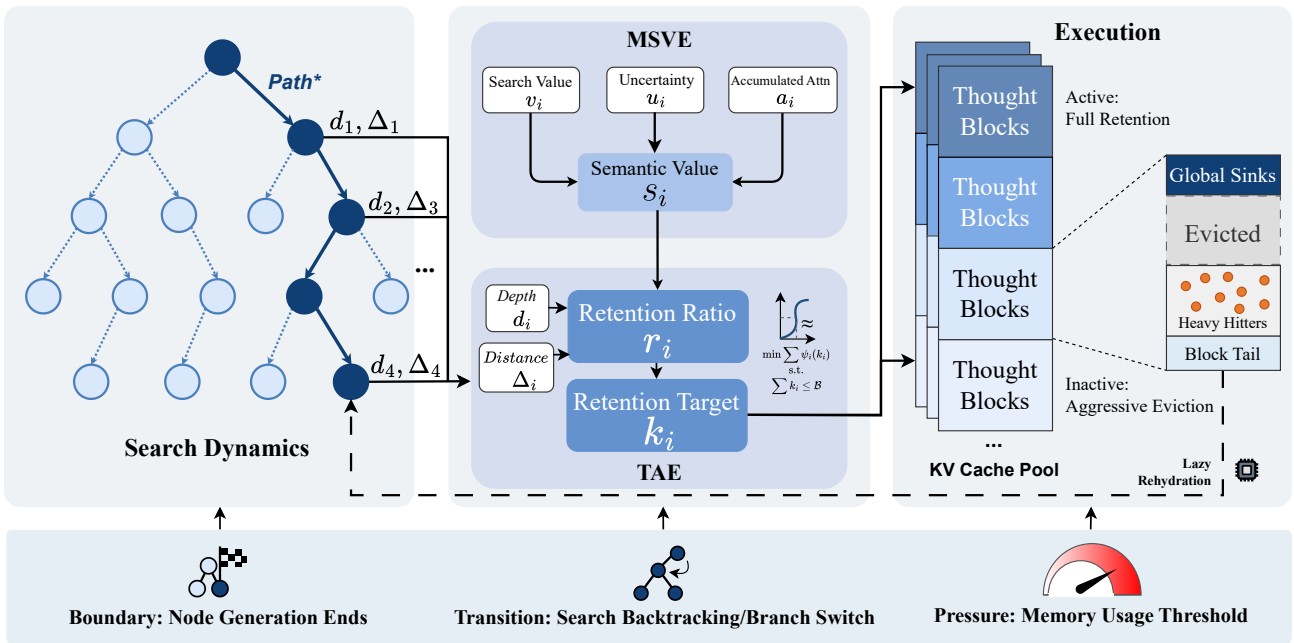

**Figure 2. Overview of the ArborKV framework.** The pipeline consists of three stages: (1) **Multi-Signal Value Estimation (MSVE)**, which fuses search value $v_i$, uncertainty $u_i$, and accumulated attention $a_i$ into a semantic score $s_i$; (2) **Tree-Aware Eviction (TAE)**, which derives the optimal retention ratio $r_i$ by solving a budgeted allocation problem via KKT conditions, considering both $s_i$ and tree geometry (depth $d_i$ and distance $\Delta_i$); and (3) **Storage & Execution**, which performs token-extractive eviction (retaining global sinks, heavy hitters, and local tails) with **lazy rehydration** via a single prefill pass to restore KV states during backtracking.

the prospective utility of each thought block with a *Tree-Aware Eviction* (TAE) policy that maps scores and geometry to retention under a global memory budget. The design separates *scoring* (estimating value) from *allocation* (assigning retention), and executes eviction in a purely token-*extractive* manner.

**Multi-Signal Value Estimation (MSVE).** For each closed block $i$, we compute a lightweight feature vector $\phi_i$ from signals already available during decoding:

- *Search value* $v_i \in [0, 1]$ (Ding et al., 2025): the normalized node score maintained by the controller's selection score for node $i$, i.e., the quantity used to prioritize which node to expand next.

- *Uncertainty* $u_i$: normalized next-token entropy at boundary $b_i$ as in Eq. 1.

- *Accumulated attention* $a_i$(Zhang et al., 2023): the running mass of attention paid *to* tokens of block $i$ by later tokens, aggregated over a thin slice of heads/layers (maintained incrementally).

We quantify uncertainty by the normalized predictive entropy of the next-token distribution at the block boundary. Let $p_i(\cdot) = p(x_{b_i+1} = \cdot \mid x_{1:b_i})$ be the softmax distribution for the next token after closing block $i$, and let $\mathcal{V}$ be the

vocabulary. We define

$$
\begin{aligned}
H_i &= -\sum_{w \in \mathcal{V}} p_i(w) \log p_i(w), \\
u_i &= 1 - \frac{H_i}{\log |\mathcal{V}|} \in [0, 1].
\end{aligned}
\tag{1}
$$

where $u_i$ is a confidence score (higher is more confident). In practice, $H_i$ is computed from the same forward pass that produces $p_i$ (no additional decoding or sampling); for efficiency we optionally restrict the sum to the top-$K$ logits and aggregate the remaining mass into an "other" bucket.

MSVE outputs a normalized score

$$
s_i = \text{clip}\big(\sigma(\theta^\top \phi_i),\, 0,\, 1\big) \in [0, 1].
$$

We calibrate $\theta$ offline using hindsight interventions on full-retention trajectories; the calibration protocol is described in Sec. 5.1.

**Tree-Aware Eviction (TAE).** Given $s_i$ and the current tree geometry, TAE assigns a per-block retention ratio and keep-count:

$$
r_i = \text{clip}\big(\alpha\, \eta^{\mathbb{I}(i \notin \text{Path}^\star)}\, s_i^\gamma\, e^{-\lambda_d d_i}\, e^{-\lambda_\Delta \Delta_i},\, r_{\min},\, 1\big), \tag{2}
$$

where $0 < \eta \le 1$ and $\mathbb{I}(i \notin \text{Path}^\star)$ is an indicator function that equals one when block $i$ is outside the current active path and zero otherwise.

We define the mandatory tail set of block $i$ as

$$\mathcal{T}_i = \{\max(a_i, b_i - L_{\text{tail}} + 1), \ldots, b_i\},$$

so that $|\mathcal{T}_i| = \min\{L_{\text{tail}}, n_i\}$.

$$k_i = \min\{n_i, \max(K_{\min}, |\mathcal{T}_i|, \lfloor r_i \, n_i \rfloor)\}. \quad (3)$$

Eqs. 2–3 encode (i) importance amplification via $\gamma > 1$; (ii) geometric prioritization via $\lambda_\Delta$ toward the active path Path$^\star$; (iii) discrete sibling coordination via the off-path discount $\eta$; and (iv) stability floors ($K_{\min}, |\mathcal{T}_i|, r_{\min}$). The geometric term $e^{-\lambda_\Delta \Delta_i}$ provides a continuous decay based on the distance from block $i$ to the active leaf, whereas $\eta$ acts as a discrete structural penalty during transition events, such as branch switching or backtracking. Its purpose is to encourage non-active subtrees to yield part of their retention budget to the newly active branch, even when their geometric distances $\Delta_i$ are similar. The depth bias is controlled by $\lambda_d$ and can be positive or negative depending on the search regime: $\lambda_d > 0$ prioritizes shallower blocks that tend to contain reusable global context, whereas $\lambda_d < 0$ favors deeper blocks closer to the current reasoning frontier; setting $\lambda_d = 0$ removes this bias when depth carries no consistent signal.

**Execution.** Given the per-block budget $k_i$ from Eq. 3, we perform token-extractive eviction by selecting an index set of retained tokens and freeing KV entries of the rest. Motivated by the empirical *attention sink* phenomenon in long-context decoding (Xiao et al., 2023b) and the observation that a small subset of tokens dominates downstream attention utility (Zhang et al., 2023; Liu et al., 2023), we retain three components:

**(i) Global sinks.** We always preserve a small prefix set $\mathcal{S}$ from the initial prompt (system/instruction tokens), which stabilizes attention beyond the training window (Xiao et al., 2023b).

**(ii) Block tail.** For each block $i$, we preserve a mandatory tail set

$$\mathcal{T}_i = \{\max(a_i, b_i - L_{\text{tail}} + 1), \ldots, b_i\},$$

which contains the most recent $\min\{L_{\text{tail}}, n_i\}$ tokens of the block, to maintain short-range coherence for the next-step generation.

**(iii) Attention heavy hitters within the block.** For each token position $t \in \{a_i, \ldots, b_i\}$, we maintain an *accumulated attention* score

$$A_i(t) = \sum_{u > b_i} \sum_{l \in \mathcal{L}} \sum_{h \in \mathcal{H}} \text{Attn}_{u \to t}^{(l,h)},$$

computed incrementally from attention weights already materialized during decoding (using a thin slice of layers/heads

$\mathcal{L}, \mathcal{H}$ for efficiency). We then select $\mathcal{H}_i$ as the top-$m_i$ positions in block $i$ under $A_i(t)$, where $m_i = k_i - |\mathcal{T}_i|$ after reserving the mandatory tail. This realizes the "recent + heavy hitters" retention pattern (Zhang et al., 2023) without additional forward passes.

The final retained set for block $i$ is $\mathcal{R}_i = \mathcal{T}_i \cup \mathcal{H}_i$ (plus global sinks $\mathcal{S}$ shared across all blocks). When $|\mathcal{R}_i| > k_i$ due to overlaps or constraints, we prioritize tail tokens first and then heavy hitters by $A_i(t)$.

**Lazy rehydration.** Eviction removes KV states but never discards the underlying token span $x_{a_i:b_i}$ recorded in the thought tree. When backtracking makes an evicted block $i$ re-enter the active path Path$^\star$, we *lazily* restore its full KV state *only at that time* by running a single prefill pass over $x_{a_i:b_i}$ (no token generation) before the next decoding step. This yields the same conditioning state as full retention on Path$^\star$, while avoiding recomputation for blocks that are never revisited.

The exact computation of MSVE scores (Eq. 1), TAE retention (Eqs. 2–3), and the token-extractive retention rule is summarized in Appendix A.1.

**Event-driven updates.** The policy is evaluated only at three engine events to minimize overhead: BOUNDARY (a block closes), TRANSITION (the active leaf switches due to branching/backtracking), and PRESSURE (total KV usage approaches the budget $\mathcal{B}$). At each event, we recompute $k_i$ via Eqs. 2–3 and apply evictions on non-active subtrees; transition-style refocusing is aligned with tree-search engines that explicitly manage search focus and switching (Ding et al., 2025).For completeness, the full event-handling logic for BOUNDARY, TRANSITION, and PRESSURE (including active-path pinning and lazy rehydration) is provided in Appendix A.2.

**Optimization view.** TAE can be interpreted as solving a separable, budgeted allocation problem on the current search tree. Let $w_i \triangleq s_i^\gamma \exp(-\lambda_d d_i) \exp(-\lambda_\Delta \Delta_i)$ denote the *value–geometry weight* of block $i$, and let $k_i \in (0, n_i]$ be the number of retained tokens. Ignoring clipping and hard floors momentarily, consider the convex program

$$\min_{\{k_i\}} \sum_{i \in V} \psi_i(k_i) \quad \text{s.t.} \quad \sum_{i \in V} k_i \leq \mathcal{B}, \ \ 0 < k_i \leq n_i,$$

$$\psi_i(k) \triangleq -w_i \log k.$$

The objective is separable and convex, and $\psi_i$ yields a diminishing-return marginal benefit $\psi_i'(k) = -w_i/k$. The KKT stationarity condition gives $-w_i/k_i^\star + \lambda = 0$ for non-saturated blocks, hence $k_i^\star = w_i/\lambda$. Accounting for the

upper bounds $k_i \leq n_i$ yields the familiar *capped* allocation

$$k_i^\star = \min\left\{n_i, \ \frac{w_i}{\lambda}\right\},$$

where the dual variable $\lambda$ (equivalently a global scale $\alpha \propto 1/\lambda$) is chosen so that $\sum_i k_i^\star = \mathcal{B}$. Expressed as a retention ratio, this gives $r_i^\star = k_i^\star / n_i \propto w_i$ up to the global scale, matching the multiplicative structure in Eq. 2. Finally, $\text{clip}(\cdot)$ and the hard floors in Eq. 3 can be viewed as projecting this relaxed solution onto a stricter feasible set: enforcing $0 \leq r_i \leq 1$ and injecting stability constraints (minimum context per block and tail preservation) that are intentionally not modeled in the relaxed objective.

## 5. Experiments

This section evaluates whether **ArborKV** improves the budget–quality–efficiency trade-off in *tree-structured* reasoning, where branching and backtracking substantially amplify KV-cache footprint. We report (i) solution quality under explicit memory budgets, (ii) end-to-end efficiency under realistic tree-search execution, and (iii) component-level ablations that connect empirical behavior to our design choices in MSVE/TAE and intra-block token selection.

### 5.1. Experimental Setup

**Hardware and runtime.** All experiments are conducted on a single NVIDIA RTX 4090 GPU (24 GB) with FP16 inference. Unless stated otherwise, we run tree search with batch size 1 and log three complementary measurements: (i) *maximum CUDA memory allocated* to capture peak device pressure, (ii) *peak cached tokens* as a hardware-stable proxy of KV footprint across kernels and allocator effects, and **(iii) average end-to-end latency** to quantify the runtime overhead introduced by event-driven control and lazy rehydration. In the main text, Table 1 reports (i) as "Peak" (GiB) after resetting CUDA memory statistics at the beginning of each episode; we defer (ii) and (iii) as well as rehydration breakdowns to Appendix C.1 for completeness.

**Models.** We evaluate two open-weight instruction-tuned LLMs practical on a 24 GB GPU: **Llama-3.1-8B-Instruct** (Dubey et al., 2024) and **Qwen2.5-7B-Instruct** (Bai et al., 2025). Decoding settings are kept identical across methods (temperature 0.7, top-$p$ 0.9; maximum generation is controlled by the search budget) so that differences arise from KV retention rather than sampling. Unless otherwise specified, all methods share the same prompts and controller/verifier logic so that the only degree of freedom is the cache policy.

**MSVE calibration.** We calibrate the MSVE parameter $\theta$ offline before inference. For each dataset, we collect 100 full-retention trajectories using the same prompts, controller,

verifier, and decoding settings as in the main experiments. For each trajectory, we perform leave-one-out masking at the thought-block level: the KV cache corresponding to each closed thought block $i$ is individually zeroed out, and the resulting change in answer accuracy is measured. This accuracy drop is used as the supervision signal for optimizing $\theta$, so that blocks whose removal causes larger performance degradation receive higher MSVE targets. After calibration, $\theta$ is fixed during inference and introduces no additional online forward passes.

**Tasks.** We focus on ToT-style reasoning workloads with explicit branching/backtracking: **GSM8K** (Cobbe et al., 2021), **MATH500** (Hendrycks et al., 2021), **Game of 24** as used in Tree-of-Thoughts (Yao et al., 2023), and **AIME** (Finkelstein et al., 2024). For rapid iteration we use a development subset of 200 instances per task when applicable; we additionally provide full-split evaluations and extended statistics in the final release materials to facilitate direct comparison with established benchmarks. We follow the standard task evaluation protocols used by prior work for these benchmarks, and we keep the controller/verifier and decoding hyperparameters fixed to avoid confounding improvements from prompt or sampling changes.

**Tree-search configurations.** We use two settings to stress different failure modes of KV management. **Config-S (budget sweep)** fixes the search shape and varies memory: branching factor $B=3, D=6, N_{\text{expand}}=64, T_{\text{node}}=128$. **Config-L (fixed-memory scaling)** increases branching and expansion to test feasibility under tight budgets: $B=5, D=8, N_{\text{expand}}=256, T_{\text{node}}=128$. All cache policies share the same ToT controller/verifier; only KV retention differs. **The controller is built on a DPTS-style transition mechanism (Ding et al., 2025) to enable efficient parallelized search.** This design ensures that differences in quality and efficiency can be attributed to the cache policy rather than changes in search order or expansion heuristics.

**Baselines.** We compare ArborKV against representative *sequence-centric* KV management methods—**H2O** (Zhang et al., 2023), **StreamingLLM** (Xiao et al., 2023b), and **ThinKV** (Ramachandran et al., 2025)—which estimate token-level utility for retention/eviction without explicitly modeling tree topology. For a controlled comparison under tree search, all methods share the same ToT controller/verifier and identical decoding hyperparameters; they differ only in how KV states are retained under the same normalized budget $\rho$. Since these baselines are designed for linear decoding, we adapt them to ToT by treating the *currently active search path* as the streaming sequence: whenever the controller expands a child or backtracks to an ancestor, we recompute the baseline's retained-token set on the concatenated tokens along that active path and evict KV entries

accordingly, while keeping the controller and expansion order unchanged. We report lazy rehydration only for ArborKV, because reversible restoration requires tree-level ownership of evicted thought-block spans, which sequence-centric baselines do not define. These baselines therefore follow their native irreversible eviction semantics: evicted tokens are permanently unavailable within an episode. To disentangle the contribution of the allocation policy from the recovery mechanism, we additionally evaluate an ArborKV variant with lazy rehydration disabled in Appendix C.3.

**Budgeting and reported statistics.** We report results under a normalized KV budget ratio $\rho$, where smaller $\rho$ corresponds to a tighter global KV waterline (cf. the budgeted formulation in Sec. 3). For clarity, the primary notion of "matched memory" in this work refers to evaluation under the same normalized budget $\rho$; the reported peak memory in GiB (maximum CUDA allocated) may vary slightly across methods due to allocator effects, kernel workspaces, and rehydration-related transient buffers. In addition to quality and efficiency, we report *rehydration count* ("Rehyd") for ArborKV, which measures how often evicted blocks must be restored when they re-enter the active path; this serves as a concrete indicator of backtracking-induced recovery and complements the latency measurements.

### 5.2. Main Results: Budget Sweep (Config-S)

Table 1 summarizes the budget–quality–efficiency trade-off across four reasoning benchmarks: **GSM8K**, **MATH500**, **Game of 24**, and **AIME**, using two representative LLMs: Llama-3.1-8B and Qwen-2.5-7B. The core question is whether a structure-aware policy can preserve reasoning-critical context under aggressive budgets *without* collapsing to the sequence-centric "flattening" behavior that over-penalizes inactive branches.

Across all models, tasks, and budgets, **ArborKV consistently outperforms sequence-centric baselines** under the same normalized budget $\rho$. While Table 1 reports peak CUDA memory allocated as "Peak" (GiB), our conclusions are driven by a controlled budget sweep in $\rho$; we additionally provide peak cached tokens, latency, and rehydration statistics in Appendix C.1 to further validate that improvements are not explained by unaccounted memory usage or hidden compute trade-offs. On **GSM8K**, ArborKV retains a large fraction of FullKV quality at $\rho = 0.50$ while substantially reducing peak memory, indicating that topology-aware retention can preserve reasoning-critical context even when only a limited portion of the cache is kept active. On the more diverse **MATH500** benchmark, ArborKV shows a similarly favorable degradation curve as budgets tighten, suggesting robustness beyond short-form arithmetic word problems and improved stability when the search encounters heterogeneous solution paths.

The advantage of ArborKV is most pronounced in backtracking-heavy workloads. On the search-intensive **Game of 24**, sequence-centric methods suffer substantial reasoning failures under frequent branch switching due to irreversible eviction on the active path, whereas ArborKV aligns allocation with search dynamics and supports recovery via lazy rehydration when previously evicted branches become relevant again. On the hard **AIME** split, ArborKV remains competitive under tight budgets, serving as a stress test that highlights the necessity of topology-aware retention for scaling tree search under fixed hardware constraints. Detailed statistics on inference latency and rehydration counts are provided in Appendix C.1, including an analysis of when rehydration is triggered and how its frequency correlates with backtracking intensity.

### 5.3. Fixed-Memory Search Scaling

We next evaluate whether ARBORKV's memory savings translate into *search scaling* under a fixed memory budget ($\rho = 0.25$). Compared to the standard configuration, this setting increases branching and expansion factors, significantly stressing peak KV allocation.

As shown in our scaling experiments, the baseline FULLKV triggers an Out-of-Memory (OOM) error when the expansion factor reaches $N_{\text{expand}} = 256$, illustrating the practical constraints of full KV retention in long-horizon reasoning. In contrast, ARBORKV successfully completes the high-expansion search without exceeding the budget, achieving $78.4\%$ accuracy with only $5.6$ GiB of peak memory. Furthermore, ARBORKV maintains competitive end-to-end latency ($38.2$s) and outperforms sequence-centric baselines at the same expansion scale, demonstrating that structure-aware retention can unlock qualitatively larger feasible search on fixed hardware.

A concise summary of scaling feasibility is reported in Table 6, while detailed resource usage comparisons (Table 2) are provided in Appendix C.2.

### 5.4. Ablations

We perform ablations in the most memory-constrained setting ($\rho = 0.25$, Config-S) to isolate which design choices drive the gains. Overall, the results suggest three takeaways: (i) MSVE is most effective when combining complementary signals rather than relying on a single proxy, (ii) tree-aware allocation remains important beyond per-block scoring, and (iii) explicitly modeling topological proximity to the active frontier is critical for both solution quality and rehydration efficiency. In addition to end-to-end accuracy, we report cache-level metrics (e.g., rehydration counts) to disentangle improvements from faster recovery versus better retention decisions. Table 3 summarizes the key component ablations; additional intra-block analyses and MSVE transferability

**Table 1.** Main Results (Config-S): Accuracy (%) and Peak Memory (GiB) across four tasks using Llama-3.1-8B and Qwen-2.5-7B. **ArborKV** reduces memory footprint under budgeted KV retention while maintaining high reasoning fidelity.

| Model | Method | $\rho$ | GSM8K | | MATH500 | | Game of 24 | | AIME | |
|---|---|---|---|---|---|---|---|---|---|---|
| | | | Acc (%) | Peak | Acc (%) | Peak | Acc (%) | Peak | Acc (%) | Peak |
| Llama-3.1-8B | FullKV | 1.00 | 80.1 | 21.8 | 79.8 | 23.5 | 56.4 | 22.4 | 28.3 | 23.0 |
| | H2O | 0.50 | 73.9 | 10.6 | 72.2 | 12.4 | 44.8 | 11.4 | 25.6 | 10.2 |
| | StreamingLLM | 0.50 | 76.1 | 11.4 | 74.2 | 13.1 | 45.6 | 12.1 | 26.1 | 13.5 |
| | ThinKV | 0.50 | 75.3 | 11.1 | 73.8 | 11.5 | 47.2 | 11.5 | 26.5 | 11.2 |
| | **ArborKV** | 0.50 | 77.8 | 11.0 | 75.8 | 12.3 | 49.1 | 11.1 | 27.2 | 11.5 |
| | H2O | 0.25 | 65.6 | 5.1 | 60.2 | 6.0 | 38.6 | 5.8 | 20.5 | 4.9 |
| | StreamingLLM | 0.25 | 63.1 | 5.7 | 62.4 | 5.8 | 36.1 | 5.4 | 21.6 | 6.1 |
| | ThinKV | 0.25 | 67.1 | 5.3 | 61.6 | 5.4 | 41.5 | 5.6 | 23.1 | 6.2 |
| | **ArborKV** | 0.25 | 70.2 | 5.5 | 64.8 | 5.9 | 44.8 | 5.8 | 24.3 | 6.1 |
| Qwen-2.5-7B | FullKV | 1.00 | 89.6 | 21.5 | 81.2 | 23.1 | 64.2 | 20.3 | 35.6 | 22.5 |
| | H2O | 0.50 | 82.0 | 12.1 | 74.8 | 10.7 | 52.7 | 11.2 | 31.5 | 10.3 |
| | StreamingLLM | 0.50 | 84.6 | 12.3 | 76.4 | 11.6 | 53.4 | 11.3 | 31.9 | 11.4 |
| | ThinKV | 0.50 | 84.0 | 10.2 | 76.0 | 10.6 | 55.0 | 10.6 | 33.2 | 11.6 |
| | **ArborKV** | 0.50 | 86.2 | 10.5 | 77.6 | 10.9 | 59.9 | 10.4 | 33.8 | 11.6 |
| | H2O | 0.25 | 71.5 | 5.3 | 61.6 | 5.4 | 42.4 | 5.3 | 19.6 | 5.3 |
| | StreamingLLM | 0.25 | 70.1 | 5.7 | 62.8 | 5.9 | 41.9 | 4.7 | 22.4 | 5.3 |
| | ThinKV | 0.25 | 69.1 | 5.6 | 65.4 | 6.0 | 49.3 | 5.7 | 25.0 | 5.7 |
| | **ArborKV** | 0.25 | 74.9 | 6.0 | 67.4 | 6.1 | 52.6 | 4.8 | 26.1 | 5.5 |

**Table 2.** Scaling Resources: Peak memory and latency under Config-L.

| Method | $N_{\text{expand}}$ | Peak (GiB) | Time (s) |
|---|---|---|---|
| FullKV | 64 | 5.9 | 18.5 |
| FullKV | 128 | 8.7 | 32.2 |
| FullKV | 256 | – | – |
| H2O | 256 | 5.6 | 42.5 |
| StreamingLLM | 256 | 5.7 | 41.8 |
| **ArborKV** | 256 | 5.6 | **38.2** |

results are deferred to Appendix C.3.

**Table 3.** Ablation - Signals: Impact of MSVE and TAE components on accuracy and efficiency.

| Variant ($\rho$=0.25) | Acc (%) | Time (s) | Rehyd |
|---|---|---|---|
| **ArborKV (full)** | **70.2** | **21.5** | 2.7 |
| w/o $v_i$ (Search Value) | 68.8 | 22.1 | 2.9 |
| w/o $u_i$ (Uncertainty) | 67.5 | 21.6 | 2.6 |
| w/o $a_i$ (Attention) | 68.1 | 22.5 | 2.5 |
| w/o sibling coord. | 69.1 | 21.9 | 2.7 |
| TAE only (no MSVE) | 65.5 | 23.4 | 3.1 |
| MSVE only (no $\Delta_i$) | 59.2 | 28.5 | 4.8 |

In particular, removing any individual MSVE signal ($v_i$, $u_i$, or $a_i$) leads to a consistent accuracy drop, indicating that the fused estimator is more robust than any single heuristic. Disabling sibling coordination produces a smaller yet

systematic degradation, suggesting that local competition among siblings complements global prioritization. We further evaluate whether the calibrated MSVE weights overfit to a specific dataset in Appendix C.3, where a GSM8K-calibrated scorer transfers well to other mathematical reasoning tasks and remains close to the default task-calibrated ArborKV.

Most notably, removing the geometric distance term (MSVE-only; no $\Delta_i$) yields the largest quality loss and substantially increases rehydration. This supports our central motivation: during tree search, short-term reuse is primarily governed by *topological proximity* to the active frontier. Ignoring this structure tends to over-evict blocks that soon become relevant again, inflating recovery work and degrading reasoning fidelity.

To separate the contribution of reversible restoration from the allocation policy itself, we additionally disable lazy rehydration while keeping the same MSVE scoring and TAE allocation policy. In this variant, once a block is evicted, it is treated as permanently unavailable within the episode, matching the irreversible eviction semantics of sequence-centric baselines. As shown in Table 4, ArborKV without rehydration still outperforms ThinKV under the same budget, while full ArborKV performs best. This indicates that the

**Table 4.** Ablation of lazy rehydration at $\rho = 0.25$. ArborKV without rehydration keeps the same MSVE and TAE policy but disables recovery of evicted blocks during backtracking.

| Task | Model | ThinKV | No Rehyd. | Full |
|------|-------|--------|-----------|------|
| GSM8K | Llama-3.1-8B | 67.1 | 68.6 | 70.2 |
|       | Qwen-2.5-7B  | 69.1 | 72.1 | 74.9 |
| MATH500 | Llama-3.1-8B | 61.6 | 63.4 | 64.8 |
|         | Qwen-2.5-7B  | 65.4 | 66.5 | 67.4 |

gains come from both tree-aware allocation and reversible restoration, rather than from rehydration alone.

We also ablate the intra-block token-retention rule in Appendix C.3. Purely recency-based selection (tail-only) performs worst, while incorporating within-block heavy hitters achieves the best performance under the same peak memory budget, supporting a "recent + salient" retention pattern as an effective token-extractive approximation of within-block utility.

### 5.5. Efficiency Breakdown

We finally examine the runtime implications of policy control and recovery *under the same setup as Sec. 5.1* (FP16 inference on a single RTX 4090 with batch size 1; identical ToT controller/verifier and decoding hyperparameters across methods, differing only in KV policy). We instrument the engine to separately time (i) event-driven policy updates (MSVE scoring + TAE allocation) triggered only at BOUNDARY/TRANSITION/PRESSURE events, and (ii) lazy rehydration, implemented as a single-pass *prefill* that restores evicted KV states without autoregressive generation. Aggregated over the Config-S budget-sweep runs in Table 1, policy control accounts for $< 1.2\%$ of total wall-clock time. Rehydration remains a second-order term: single-pass prefilling is substantially faster than generating the same tokens, and the moderate rehydration counts in Table 5 indicate that most evicted blocks are not immediately revisited.

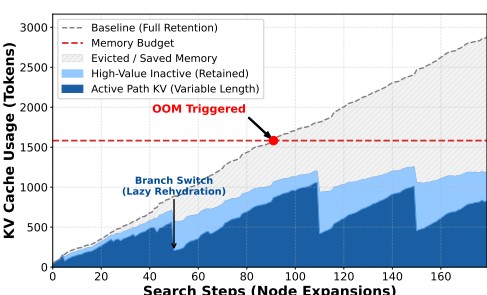

**Figure 3. KV memory dynamics. ArborKV** avoids the **OOM** failure of the full-retention baseline (grey) by dynamically retaining only the **Active Path** (dark blue) and essential **Inactive** context (light blue). Sharp drops illustrate structure-aware eviction upon branch switching.

## 6. Conclusion

We propose **ArborKV**, a tree-aware KV-cache management framework for inference-time tree search with large language models. By coupling semantic value estimation with tree-structured budget allocation, ArborKV enables reversible exploration under strict memory constraints via token-extractive eviction and lazy rehydration. Experiments on Tree-of-Thoughts style reasoning tasks show that ArborKV reduces KV memory and improves efficiency while preserving reasoning performance, underscoring the value of modeling search structure beyond linear decoding.

## Acknowledgment

This research is supported by Smart-Grid National Science and Technology Major Project (Grant No. 2025ZD0805500).

## Impact Statement

This work targets the systems bottleneck of tree-structured LLM inference by reducing the key–value (KV) cache footprint under fixed device memory budgets. By enabling larger or longer-horizon Tree-of-Thoughts–style search on a single commodity GPU, the proposed approach may lower the hardware and energy cost of advanced reasoning pipelines and broaden access to such techniques in resource-constrained settings.

At the same time, improving the efficiency of deliberative search could also increase the scale at which LLM reasoning is deployed, potentially amplifying existing risks of misuse (e.g., generating persuasive but incorrect reasoning traces) and over-reliance on model outputs. Our method does not introduce new training data, does not change model weights, and operates only on inference-time cache management; therefore it does not create additional privacy exposure beyond standard inference. We encourage responsible deployment practices, including task-appropriate verification, monitoring for failure modes under aggressive eviction budgets, and adherence to applicable safety policies when scaling automated reasoning systems.

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

**Algorithm 1** MSVE + TAE + Token-Extractive Eviction (Bundled Params)

1: **Procedure** $\text{MSVE}(v_i, u_i, a_i; \Pi)$
2:     $\phi_i \leftarrow [v_i, u_i, a_i]$
3:     **return** $\text{clip}(\sigma(\theta(\Pi)^\top \phi_i), 0, 1)$
4: **Procedure** $\text{TAE}(s, d, \Delta, n; \Pi)$
5:     $z \leftarrow \alpha(\Pi)\, s^{\gamma(\Pi)} e^{-\lambda_d(\Pi)\, d} e^{-\lambda_\Delta(\Pi)\, \Delta}$
6:     $r \leftarrow \text{clip}(z, r_{\min}(\Pi), 1)$
7:     $k \leftarrow \max\{K_{\min}(\Pi), L_{\text{tail}}(\Pi), \lfloor rn \rfloor\}$
8:     **return** $\min\{k, n\}$
9: **Procedure** $\text{EVICT}(i, k_i; \Pi, \mathcal{S})$
10:    $k_i \leftarrow \min\{k_i, n_i\}$
11: **if** $k_i \leq L_{\text{tail}}(\Pi)$ **then**
12:     **return** $\{b_i - k_i + 1, \ldots, b_i\} \cup \mathcal{S}$
13: **end if**
14:    $\mathcal{T}_i \leftarrow \{b_i - L_{\text{tail}}(\Pi) + 1, \ldots, b_i\}$
15:    $m_i \leftarrow k_i - |\mathcal{T}_i|$
16:    $\mathcal{H}_i \leftarrow \text{TOP-}m_i \text{ BY } A_i(t)$
17:    **return** $\text{TRIM}(\mathcal{T}_i, \mathcal{H}_i, k_i) \cup \mathcal{S}$
18: **Procedure** $\text{SCOREALLOCEVICT}(i; \Pi, \mathcal{S})$
19:    $s_i \leftarrow \text{MSVE}(v_i, u_i, a_i; \Pi)$
20:    $k_i \leftarrow \text{TAE}(s_i, d_i, \Delta_i, n_i; \Pi)$
21:    $\mathcal{R}_i \leftarrow \text{EVICT}(i, k_i; \Pi, \mathcal{S})$
22:    **return** $(s_i, k_i, \mathcal{R}_i)$

## A. Additional Pseudocode and Implementation Details

### A.1. MSVE scoring, TAE allocation, and token-extractive eviction

Algorithm 1 summarizes the three-stage policy evaluation: MSVE scoring, TAE allocation, and token-extractive eviction. MSVE computes $s_i$ from $(v_i, u_i, a_i)$; TAE maps $(s_i, d_i, \Delta_i)$ to $(r_i, k_i)$ following Eqs. 2–3; eviction retains global sinks $\mathcal{S}$, the tail window $\mathcal{T}_i$, and within-block attention heavy hitters $\mathcal{H}_i$, prioritizing tail tokens under overlaps.

### A.2. Event-driven controller

We provide pseudocode for the event-driven KV-cache controller used by our tree-search inference engine. The controller updates retention targets only at three policy update events (BOUNDARY, TRANSITION, PRESSURE), enforces active-path protection, and performs lazy rehydration via a single-shot prefill when a previously evicted block re-enters Path$^\star$. Algorithm 2 lists the complete control flow.

## B. More Related Work

### B.1. LLM Reasoning and Post-Training

Recent LLMs have shown stronger reasoning ability through both prompting-time and training-time improvements. Chain-of-thought prompting and self-consistency

**Algorithm 2** Event-Driven Controller (PUEs + Rehydration)

**Require:** Budget $\mathcal{B}$; parameter bundle $\Pi$; sinks $\mathcal{S}$
**Ensure:** $\{k_i\}_{i \in V}$ and retained sets $\{\mathcal{R}_i\}_{i \in V}$
1: **while** tree-search running **do**
2:    $e \leftarrow \text{NEXTPUE}()$
3:    **if** $e = \text{BOUNDARY}(i)$ **then**
4:      $(s_i, k_i, \mathcal{R}_i) \leftarrow \text{SCOREALLOCEVICT}(i; \Pi, \mathcal{S})$
5:    **else if** $e = \text{TRANSITION}(\ell_{\text{new}})$ **then**
6:      $\ell^\star \leftarrow \ell_{\text{new}}$
7:      $\text{Path}^\star \leftarrow \text{ROOTTOLEAF}(\ell^\star)$
8:      **for all** $i \in \text{Path}^\star$ **do**
9:        **if** $k_i < n_i$ **then**
10:         $\text{REHYDRATE}(x_{a_i:b_i})$
11:         $k_i \leftarrow n_i$
12:         $\mathcal{R}_i \leftarrow \{a_i, \ldots, b_i\} \cup \mathcal{S}$
13:        **end if**
14:      **end for**
15:      **for all** $j \in V \setminus \text{Path}^\star$ **do**
16:        $\Delta_j \leftarrow \text{TREEDISTANCE}(j, \ell^\star)$
17:        $k_j^{\text{new}} \leftarrow \text{TAE}(s_j, d_j, \Delta_j, n_j; \Pi)$
18:        **if** $k_j^{\text{new}} < k_j$ **then**
19:         $k_j \leftarrow k_j^{\text{new}}$
20:         $\mathcal{R}_j \leftarrow \text{EVICT}(j, k_j; \Pi, \mathcal{S})$
21:        **end if**
22:      **end for**
23:    **else if** $e = \text{PRESSURE}$ **then**
24:      **for all** $j \in V \setminus \text{Path}^\star$ **do**
25:        $k_j \leftarrow \text{TAE}(s_j, d_j, \Delta_j, n_j; \Pi)$
26:        $\mathcal{R}_j \leftarrow \text{EVICT}(j, k_j; \Pi, \mathcal{S})$
27:      **end for**
28:      **while** $\sum_{i \in V} k_i > \mathcal{B}$ **do**
29:        $j \leftarrow \arg\min_{i \in V \setminus \text{Path}^\star} \text{PRIORITY}(i)$
30:        $k_j \leftarrow \max\{K_{\min}(\Pi), k_j - 1\}$
31:        $\mathcal{R}_j \leftarrow \text{EVICT}(j, k_j; \Pi, \mathcal{S})$
32:      **end while**
33:    **end if**
34:    **if** $\sum_{i \in V} k_i \geq \mathcal{B} - \delta(\Pi)$ **then**
35:      $\text{RAISEPRESSUREEVENT}()$
36:    **end if**
37: **end while**

improve multi-step reasoning along linear trajectories, while tree-structured methods further introduce explicit branching and search over intermediate thoughts (Wei et al., 2022; Wang et al., 2022; Yao et al., 2023). In parallel, instruction tuning and reinforcement learning from human feedback have become important post-training tools for aligning model behavior (Ouyang et al., 2022). Recent studies also explore step-level faithfulness and reasoning-length optimization, aiming to make reasoning more reliable and efficient (Gui et al., 2026a;b). ArborKV is orthogonal to these advances, as it improves the memory efficiency of tree-based reasoning at inference time without changing the

**Table 5.** Detailed Efficiency Statistics (Config-S): Inference Time (s) and Rehydration Count (Rehyd) across four tasks. Rehydration is a unique mechanism in ARBORKV to support lazy restoration during backtracking.

| Model | Method | $\rho$ | GSM8K | | MATH500 | | Game of 24 | | AIME | |
|---|---|---|---|---|---|---|---|---|---|---|
| | | | Time (s) | Rehyd | Time (s) | Rehyd | Time (s) | Rehyd | Time (s) | Rehyd |
| | FullKV | 1.00 | 69.7 | 0.0 | 190.6 | 0.0 | 98.4 | 0.0 | 145.1 | 0.0 |
| | H2O | 0.50 | 24.1 | – | 95.3 | – | 46.5 | – | 66.5 | – |
| | StreamingLLM | 0.50 | 23.8 | – | 77.6 | – | 34.2 | – | 78.2 | – |
| | ThinKV | 0.50 | 25.5 | – | 54.9 | – | 39.2 | – | 61.3 | – |
| Llama-3.1-8B | **ArborKV** | 0.50 | 19.8 | 0.7 | 51.2 | 0.9 | 31.4 | 1.9 | 59.7 | 1.3 |
| | H2O | 0.25 | 22.4 | – | 125.6 | – | 32.1 | – | 72.6 | – |
| | StreamingLLM | 0.25 | 22.1 | – | 89.3 | – | 31.5 | – | 79.1 | – |
| | ThinKV | 0.25 | 23.6 | – | 53.2 | – | 34.8 | – | 60.5 | – |
| | **ArborKV** | 0.25 | 21.5 | 2.7 | 61.2 | 3.6 | 33.7 | 5.2 | 61.6 | 3.1 |
| | FullKV | 1.00 | 79.7 | 0.0 | 125.0 | 0.0 | 112.4 | 0.0 | 95.6 | 0.0 |
| | H2O | 0.50 | 27.6 | – | 67.3 | – | 41.7 | – | 45.6 | – |
| | StreamingLLM | 0.50 | 27.2 | – | 78.9 | – | 39.1 | – | 57.3 | – |
| | ThinKV | 0.50 | 29.1 | – | 72.6 | – | 44.8 | – | 42.1 | – |
| Qwen-2.5-7B | **ArborKV** | 0.50 | 22.6 | 0.8 | 61.2 | 1.3 | 35.9 | 2.3 | 43.9 | 0.9 |
| | H2O | 0.25 | 25.6 | – | 57.8 | – | 36.6 | – | 47.3 | – |
| | StreamingLLM | 0.25 | 25.2 | – | 72.1 | – | 35.9 | – | 51.5 | – |
| | ThinKV | 0.25 | 26.9 | – | 60.9 | – | 39.8 | – | 47.6 | – |
| | **ArborKV** | 0.25 | 24.5 | 2.4 | 52.5 | 2.5 | 40.1 | 4.7 | 44.3 | 3.4 |

**Table 6.** Scaling Performance (Config-L, $\rho = 0.25$) on **MATH500** using **Llama-3.1-8B**: Accuracy and OOM status across $N_{\text{expand}}$.

| Method | $N_{\text{expand}}$ | Acc (%) | OOM |
|---|---|---|---|
| FullKV | 64 | 79.8 | N |
| FullKV | 128 | 80.5 | N |
| FullKV | 256 | – | Y |
| H2O | 256 | 69.0 | N |
| StreamingLLM | 256 | 67.1 | N |
| **ArborKV** | 256 | **78.4** | N |

**Table 7.** Ablation (Config-S, $\rho = 0.25$) on **GSM8K** using **Llama-3.1-8B**: intra-block token-selection rules under a fixed peak-memory cap (5.6 GiB).

| Intra-block Policy | Acc (%) | Peak (GiB) |
|---|---|---|
| Tail-only | 65.2 | 5.6 |
| Sinks + Tail | 68.5 | 5.6 |
| Sinks + Tail + HeavyHitters | 70.8 | 5.6 |

model or its reasoning objective.

### B.2. Efficient Inference

Efficient LLM inference has been studied from system, architecture, and cache-management perspectives, including paged KV-cache allocation, offloading, quantization, and token-level cache eviction (Kwon et al., 2023; Sheng et al., 2023; Zhang et al., 2023; Xiao et al., 2023b). Recent KV-

cache methods further refine token selection with adaptive or reasoning-aware policies (Feng et al., 2024; Ramachandran et al., 2025). Related efficiency efforts also extend to multimodal models, where visual-token sparsification or pruning can reduce the cost of vision-language and vision-language-action inference (Zhang et al., 2025; Chen et al., 2024; Liu et al., 2025). These methods mainly target linear decoding or multimodal token redundancy, whereas ArborKV focuses on tree topology and backtracking dynamics in search-based LLM reasoning.

## C. Additional Experimental Statistics

### C.1. Detailed Main Results (Config-S)

Table 5 provides additional metrics for the Budget Sweep (Config-S) experiments, including end-to-end inference time and the average rehydration count per sample. These statistics quantify the efficiency gains of ARBORKV and the overhead introduced by its lazy rehydration mechanism during non-monotonic search.

### C.2. Detailed Scaling Analysis (Config-L)

This section provides the complete experimental data for fixed-memory search scaling. A concise summary of feasibility and accuracy (including OOM status) is reported in Ta-

**Table 8.** Transferability of MSVE calibration at $\rho = 0.5$. "GSM8K-tuned" uses $\theta$ calibrated only on GSM8K and applies it zero-shot to other tasks. "Default" uses the task-specific calibration adopted in the main experiments.

| Model | Task | ThinKV | GSM8K-tuned | Default |
|---|---|---|---|---|
| Llama-3.1-8B | GSM8K | 75.3 | 77.8 | 77.8 |
| | MATH500 | 73.8 | 75.1 | 75.8 |
| | Game of 24 | 47.2 | 48.3 | 49.1 |
| | AIME | 26.5 | 26.9 | 27.2 |
| Qwen-2.5-7B | GSM8K | 84.0 | 86.2 | 86.2 |
| | MATH500 | 76.0 | 77.1 | 77.6 |
| | Game of 24 | 55.0 | 58.2 | 59.9 |
| | AIME | 33.2 | 33.5 | 33.8 |

ble 6 in the main text; here we provide detailed resource usage, including peak memory and end-to-end latency across expansion configurations (Table 2). In particular, Table 6 reports the feasibility and accuracy on MATH500 with Llama-3.1-8B under Config-L at $\rho = 0.25$.

## C.3. Additional Ablation Results

This appendix provides complementary ablation results for ArborKV under memory-constrained settings. In addition to the component-level ablations and the lazy-rehydration ablation reported in Sec. 5.4, we further examine two aspects: (i) the effect of intra-block token selection, and (ii) the transferability of MSVE calibration across mathematical reasoning tasks.

**Intra-block token selection.** Table 7 evaluates intra-block token-selection strategies under a fixed peak-memory budget. We compare tail-only retention, adding global sinks, and further incorporating within-block heavy hitters. All variants use the same ToT controller and memory budget to isolate the effect of intra-block retention. The results on GSM8K with Llama-3.1-8B under Config-S at $\rho = 0.25$ show that purely recency-based retention performs worst, while combining global sinks, block tails, and heavy hitters achieves the best performance under the same 5.6 GiB peak-memory cap.

**Transferability of MSVE calibration.** We evaluate whether the MSVE parameter $\theta$ generalizes across mathematical reasoning tasks. Specifically, we calibrate $\theta$ only on GSM8K and directly apply it to other tasks without task-specific recalibration. Table 8 shows that the GSM8K-calibrated scorer transfers well within the mathematical domain: it remains close to the default task-calibrated ArborKV and consistently outperforms ThinKV. This suggests that MSVE signals such as uncertainty $u_i$ and accumulated attention $a_i$ capture transferable reasoning patterns rather than only dataset-specific features.

