# OpenReview forum: "ArborKV: Structure-Aware KV Cache Management for Scaling Tree-based LLM Reasoning"
_ICML.cc/2026/Conference — ICML 2026 regular_

### Official Review · Reviewer_RnsK · 2026-02-18

**Soundness:** 3
**Presentation:** 3
**Significance:** 3
**Originality:** 3
**Overall Recommendation:** 4
**Confidence:** 5

**Summary:**

This paper reduces the KV cache memory consumption of Tree-of-Thoughts (ToT) style reasoning. This paper proposes ArborKV, which combines a Multi-Signal Value Estimator (MSVE) that scores thought blocks using search value, uncertainty, and attention signals, with a Tree-Aware Eviction (TAE) policy that allocates retention budgets based on topological proximity to the active search frontier. Evicted blocks can be lazily rehydrated via prefill when backtracking revisits them. Experiments on GSM8K, MATH500, Game of 24, and AIME show ~4× peak memory reduction with modest accuracy degradation.

**Compliance With Llm Reviewing Policy:**

Affirmed.

**Final Justification:**

You addressed your main concerns and made me more confident about it.

**Key Questions For Authors:**

How does θ in MSVE generalize? If calibrated on GSM8K, does it transfer to Game of 24 without re-tuning?

**Strengths And Weaknesses:**

Strengths

The motivation is clear that the memory issue is the main bottleneck of Tree-of-Thoughts (ToT), and using strategies like dynamic programming can reduce the memory pressure while maintaining the performance.

The separation of scoring (MSVE) from allocation (TAE) is principled. The optimization view in Section 4, casting TAE as a separable budgeted allocation problem with a KKT-derived closed-form solution, is elegant and provides theoretical grounding for what could otherwise seem like an ad hoc heuristic.

Weaknesses

The MSVE calibration procedure is underspecified.
The paper states that θ is calibrated offline using "hindsight interventions on full-retention trajectories: masking selected blocks yields outcome deltas that supervise θ." This is a single sentence describing what is essentially a supervised learning procedure. Key details are missing: how many trajectories are used, how masking is performed, whether θ generalizes across tasks/models, and what the sensitivity to θ is. Given that MSVE is a core contribution, this deserves substantially more detail and analysis.

The sequence-centric baselines (H2O, StreamingLLM, ThinKV) are adapted to tree search by treating the active path as a streaming sequence. Critically, these baselines do not get lazy rehydration — evicted tokens are permanently lost. This is acknowledged but creates an asymmetry: ArborKV's advantage may partly stem from the rehydration mechanism rather than the scoring/allocation policy. An ablation of ArborKV without rehydration would clarify this. Also, the paper would benefit from at least a discussion of how ArborKV interacts with model parallelism and paged attention systems like vLLM.

---

> ### Author Rebuttal · Authors · 2026-03-31
>
> Thank you for your valuable feedback! Following are our responses to each comment:
> ## Weakness about MSVE
> > MSVE calibration procedure is underspecified. How many trajectories? How masking is performed? How does θ in MSVE generalize? If calibrated on GSM8K, does it transfer?
>
> Thank you for this valuable comment! We provide the details below:
> - Trajectories&Masking: We use 100 full-retention trajectories from each dataset. For each trajectory, we perform leave-one-out masking: we individually zero out the KV cache of each thought block i and measure the resulting delta in answer accuracy. This accuracy drop serves as the supervision signal to optimize θ.
> - θ Generalization: We evaluate ArborKV (Zero-shot)—using θ calibrated on GSM8K—against the strongest sequence-centric baseline, ThinKV. The results show that θ exhibits considerable generaliability. The transferability is notable within the mathematical domain. This suggests that MSVE signals like Uncertainty ($u_i$) and Accumulated Attention ($a_i$) capture generalizable reasoning patterns rather than just dataset-specific features. For reference, we report the results below (Acc at ρ=0.5):
> |Model|Task|ThinKV|ArborKV(Tuned on GSM8K)|ArborKV(Default)
> |:-|:-|:-:|:-:|:-:
> | Llama-3.1-8B | GSM8K | 75.3 | 77.8 | 77.8
> | |MATH500| 73.8 | 75.1 | 75.8
> | |Game of 24|47.2| 48.3 | 49.1
> | |AIME|26.5|26.9|27.2
> |Qwen-2.5-7B|GSM8K|84.0|86.2|86.2
> | | MATH500 | 76.0 | 77.1 | 77.6
> | | Game of 24 | 55.0 | 58.2 | 59.9
> | | AIME | 33.2 | 33.5 | 33.8
> ## Weakness about lazy rehydration
> > The sequence-centric baselines do not get lazy rehydration. ArborKV's advantage may partly stem from this mechanism rather than scoring policy. An ablation of ArborKV without rehydration would clarify this.
>
> We appreciate this constructive suggestions!
>
> **Core Innovation:** We would like to clarify that Lazy Rehydration is one of our core architectural innovations specifically designed for tree-structured reasoning. Unlike sequence-centric baselines that assume a monolithic forward path, ArborKV recognizes that inactive subtrees must remain recoverable for backtracking. This mechanism allows to achieve memory reduction without permanent information loss on critical reasoning branches.
>
> **Ablation:** We conduct an ablation study by disabling Lazy Rehydration (ρ=0.25):
>   |Task|Model|ThinKV|ArborKV(No Rehydration)|ArborKV(Full)|
>   |:-|:-|:-:|:-:|:-:|
>   |GSM8K|Llama-3.1-8B| 67.1 | 68.6 | 70.2 |
>   ||Qwen-2.5-7B| 69.1 | 72.1 | 74.9 |
>   |MATH500|Llama-3.1-8B| 61.6 | 63.4 | 64.8 |
>   ||Qwen-2.5-7B| 65.4 | 66.5 | 67.4 |
>
> Even without lazy rehydration, ArborKV's TAE policy consistently outperforms baselines across all tasks. This confirms that our allocation policy is inherently superior for tree-structured search. While sequence-centric methods flatten the tree into a linear pool and mistakenly evict vital ancestral blocks, our TAE policy explicitly prioritizes the ancestor chain and topological proximity to the active frontier. Consequently, it protects the reasoning foundation more effectively than linear heuristics.
>
> ## Weakness about system integration
> > The paper would benefit from a discussion of how ArborKV interacts with model parallelism and paged attention systems like vLLM.
>
> We appreciate these practical comments. While production deployment requires engineering effort, integrating ArborKV into modern runtimes is architecturally feasible. It operates at the KV-cache management layer, leaving the transformer and attention mechanisms unchanged. We summarize the key points below (more details in our response to R-KDGU):
> - Compatibility: To bypass PagedAttention's token-sparsity hurdles, ArborKV supports a two-stage integration. Initially, it evaluates and unmaps entire physical pages based on MSVE scores, natively aligning with current block-level cache managers. Finer token-level compaction can be unlocked later once runtimes adopt ragged KV kernels.
> - Custom Kernels&Lazy Rehydration: Page-level operation eliminates the need for custom sparse attention kernels. Through lazy rehydration, missing pages of re-entered branches are restored via a single prefill pass, ensuring attention always processes fully populated physical pages during the decode phase.
> - Continuous Batching&Parallelism: ArborKV is event-driven (updating only at block closures or branch switches), eliminating per-token overhead during continuous batching decodes. For model parallelism, local MSVE computation on each GPU shard ensures decentralized execution with minimal synchronization.
>
> Currently, we focus on algorithmic limits. While integrating this structure-aware retention into production runtimes requires non-trivial engineering, it remains feasible and a promising direction for future work.
>
> **We will include the above analyses in the revision. We sincerely thank the reviewer again and hope these clarifications help address your concerns! If so, we would deeply appreciate it if you could consider raising your score.**

---

> > ### Author Rebuttal · Reviewer_RnsK · 2026-04-03
> >
> > We thank the authors for their response. Our concern is fully resolved. We decided to keep the score and raise my confidence to 5 as we are open to accepting this paper with big confidence.

---

> > > ### Author Response · Authors · 2026-04-03
> > >
> > > Dear Reviewer RnsK:
> > >
> > > Thank you for your kind support and for raising the confidence from 3 to 5! We sincerely appreciate your valuable suggestions. We will incorporate the discussed clarifications and additional analyses into the final version to further strengthen the paper.
> > >
> > > With gratitude,
> > >
> > > Authors

---

### Official Review · Reviewer_KDGU · 2026-03-08

**Soundness:** 4
**Presentation:** 3
**Significance:** 4
**Originality:** 4
**Overall Recommendation:** 4
**Confidence:** 5

**Summary:**

The paper studies KV-cache management under tree-structured LLM reasoning, where Tree-of-Thought (ToT) style search significantly amplifies memory usage due to branching and backtracking. The authors propose ArborKV, a structure-aware KV management framework that allocates cache budgets according to both reasoning utility and the topology of the reasoning tree.

**Compliance With Llm Reviewing Policy:**

Affirmed.

**Final Justification:**

Overall, the motivation and idea are both novel and solid. The rebuttal addressed my main concerns.

**Key Questions For Authors:**

It would be helpful if the authors could clarify how ArborKV could be integrated into modern inference runtimes

**Limitations:**

yes

**Strengths And Weaknesses:**

Strengths
1. Problem is real and underserved
The paper addresses an important systems problem. While most existing KV cache management techniques assume linear generation, tree-structured reasoning introduces new memory challenges due to the need to retain KV states for multiple partial reasoning trajectories.
2. Insightful reframing
The key insight that KV-cache allocation should be topology-aware, rather than purely sequence-centric, is compelling. Incorporating both reasoning utility and the geometric structure of the reasoning tree is a natural and promising direction.
3. Clear algorithmic structure
The separation between the value estimator (MSVE) and the allocation mechanism (TAE) makes the method conceptually clear and easy to follow.
4. Reasonably comprehensive experimental evaluation
The evaluation includes multiple reasoning benchmarks, budget sweeps, and ablations, which helps demonstrate the behavior of the proposed method under different memory constraints.
Weaknesses
While the idea is interesting, there appear to be several engineering challenges when integrating the approach into modern inference runtimes:
Token-level eviction currently not support in vllm or sglang
Sparse token retention may need custom attention kernels
Interaction with continuous batching is unclear
These issues do not invalidate the idea, but they may introduce additional engineering complexity for deployment in production inference systems.

---

> ### Author Rebuttal · Authors · 2026-03-31
>
> ## Response to Weakness&Question:
> > While the idea is interesting, there appear to be several engineering challenges when integrating the approach into modern inference runtimes: Token-level eviction currently not support in vllm or sglang Sparse token retention may need custom attention kernels Interaction with continuous batching is unclear These issues do not invalidate the idea, but they may introduce additional engineering complexity for deployment in production inference systems.
>
> > It would be helpful if the authors could clarify how ArborKV could be integrated into modern inference runtimes
>
> We thank the reviewer for this constructive comment!
>
> While an optimized production deployment entails non-trivial engineering effort, ArborKV is architecturally feasible to integrate into modern runtimes. Because it does not alter the transformer backbone or attention computation, modifications are primarily confined to extending the existing KV-cache management layer.
>
> Specifically, we address the integration concerns as follows:
>
> - Token-level eviction and vLLM/SGLang compatibility: While fine-grained token sparsity presents hurdles for standard PagedAttention, a practical integration follows a two-stage path. Initially, ArborKV can be implemented natively at the block/page granularity. Instead of evicting individual tokens, the Tree-Aware Eviction (TAE) policy can evaluate and unmap entire physical pages (e.g., 16-token blocks) based on cumulative MSVE scores and topological distances. This aligns directly with existing paged-cache managers. Finer token-extractive compaction can be unlocked later as runtimes increasingly adopt kernels supporting ragged KV caches.
>
> - Custom kernels and Lazy Rehydration: By operating at the page level for initial integration, ArborKV bypasses the need for custom sparse attention kernels. Furthermore, our lazy rehydration design naturally complements this. When an evicted branch re-enters the active path, its missing pages are restored via a single prefill pass before decoding resumes(). Consequently, the attention mechanism always operates over fully populated physical pages during the critical decode phase.
>
> - Interaction with Continuous Batching: ArborKV introduces minimal friction to continuous batching schedulers because it is strictly event-driven. Retention targets are recomputed only at specific policy update events (i.e., when a thought block closes, the search branches/backtracks, or memory pressure peaks), avoiding per-token control overhead in the decode loop. Moreover, since continuous batching concurrency is strongly bottlenecked by per-request memory, ArborKV's ability to compress the peak KV footprint directly translates to a higher feasible number of active requests.
>
> Our current implementation focuses on evaluating the algorithmic limits of tree-aware KV retention using a high-level search framework. We acknowledge that embedding these structure-aware mechanisms into production runtimes (like vLLM) needs non-trivial engineering efforts, and we consider this a promising direction for future work. We will make these engineering pathways and limitations in the revised manuscript.
>
> We sincerely thank the reviewer again and hope these clarifications help address your concerns! If so, we would deeply appreciate it if you could consider raising your score.

---

> > ### Author Rebuttal · Reviewer_KDGU · 2026-04-03
> >
> > Good idea

---

> > > ### Author Response · Authors · 2026-04-03
> > >
> > > Dear Reviewer KDGU:
> > >
> > > We greatly appreciate your constructive feedback and your willingness to raise your confidence! We will incorporate this clarification into the final version of the paper, with a clearer discussion of how ArborKV can be integrated into modern inference runtimes and the related engineering considerations.
> > >
> > > Thank you again for your valuable suggestions to improve our paper.
> > >
> > > With gratitude,
> > >
> > > Authors

---

### Official Review · Reviewer_s5J8 · 2026-03-13

**Soundness:** 3
**Presentation:** 2
**Significance:** 3
**Originality:** 3
**Overall Recommendation:** 4
**Confidence:** 3

**Summary:**

This paper addresses the KV cache problem in tree-structured LLM reasoning. As tree-structured LLM  reasoning is a rising trend, but with distinct complexities of its memory consumption on the KV cache, existing KV cache management policies are mostly targeted to LLM reasoning with single linear trajectories. This work is among the earlier efforts for tree-structured LLM reasoning, thus it  is a very important thread of work.

The framework is based on the observation that the reuse of KV cache in tree-structured LLM is governed by search and backtracking dynamics instead of simpler token-level statistics. Based on this observation/understanding, the authors introduced a tree-aware allocation principle, which combines both semantic information in reasoning and structural information (proximity). Following this principle, they proposed ArborKV, which consists of two components: one providing quantitative evaluation on thought blocks (a set tokens), the other uses this metric to the tree for providing additional information of deciding the caching/eviction policy.

The performed comprehensive empirical evaluation and compare their work against several KV management methods, the results show that this work can achieve considerable  KV memory reduction with little degradation of reasoning accuracy.

**Compliance With Llm Reviewing Policy:**

Affirmed.

**Final Justification:**

The clarifications in the rebuttal addressed my main concerns, and positively changed my evaluation.

**Key Questions For Authors:**

I have no questions.

**Limitations:**

yes

**Strengths And Weaknesses:**

Soundness:
The observation, principle, approach and framework are intuitively sounding, the experiments are comprehensive and convincing.

Presentation:
The Introduction and Related Work are well written, and the high-level approach can be clearly understood. But its Method section is hard to understand, including it notations and explanations, e.g.,
1. The explanation for Eq. 2 and 3 are intertwined. It is suggested to explain them one by one. The multiplicative discount $\eta$ appears in the explanation part, but never appear anywhere in the two equations. Also, why are $K_{min}$ and $L_{tail}$ both necessary? In fact, the meaning of $K_{min}$ has never been explained (although can be guessed), and the meaning of $L_{tail}$ appeared at a place much later.
2. On Line 253 (Page 5), $|T_i| = L_{tail}$ by definition on Line 241. So why confuse the readers with two different notations? On the other hand, according to the definition of $k_i$ in Eq. 3 on Page 3, $k_i \ge L_{tail}$, which means the max function on Line 253 is unnecessary. Also, what is the difference between $k_i$ and $L_{tail}$? According to their definitions, $k_i$ means the retained tokens for block $i$, and $L_{tail}$ is defined as the preserved most recent tokens for block $i$.
3. There are a large number of notations and terms, which seem to be not carefully considered together.

 Significance:
The tree-structured LLM reasoning is a recent and promising trend. On the other hand, this new reasoning structure has  its unique complexities that deserve in-depth investigation. This work is among the first ones studying this emerging problem, so it is considered to an impacting work.

Novelty:
From the high-level description of its technical framework, including problem understanding, observation, design principle, framework and its constituent components, it exhibits its novelty, and the empirical results also support the effectiveness of its technique. However, the technical details are not well written, including a number of confusions, this imposes negative impact on the confidence about its novelty.

---

> ### Author Rebuttal · Authors · 2026-03-31
>
> Thank you for your constructive comment! Following are our responses to each comment:
>
> >Eqs.2 and 3 are explained in an intertwined way, and $\eta$, $K_{min}$, and $L_{tail}$ are not clearly defined.
>
> **(1)Eq.2 and Eq.3**
>
> We would like to clarify that Eq.2 and Eq.3 are designed to represent a cohesive, two-stage allocation pipeline, serving distinct but sequentially dependent purposes in our algorithm. Our intention was to present them as one connected pipeline. We also agree that explaining them one by one would be clearer, and we will revise in the final version.
>
> **(2)The sibling coordination factor $\eta$**
>
> - Concept of $\eta$: $\eta$ is a discrete structural penalty applied during sibling coordination. Its purpose is to ensure that when the search focus shifts during a Transition event (branch switching or backtracking), non-active sibling subtrees immediately yield part of their budget to the newly active branch, even when their geometric distances $\Delta_i$ may be similar. $\eta$ complements the continuous geometric decay by encoding local competition among siblings.
>
> - Why it was presented separately: In our current Method section, Eqs.2 and 3 were intended to describe the base allocation policy for a given tree state, while sibling coordination was implemented as an event-driven adjustment within the Transition logic.
>
> To enhance formal rigor, the final version will integrate $\eta$ into a unified allocation formula.
>
> **(3)The meanings of $K_{min}$ and $L_{tail}$**
>
> In fact, both quantities are mentioned earlier in the Preliminaries: $K_{min}$ appears as a per-block minimum retention floor, and $L_{tail}$ is introduced as the preserved recent-token window.
>
> Their roles are in fact different and not redundant. $K_{min}$ is a minimum retention floor on the number of tokens kept for each block, preventing a block from being overly emptied when its score is low. $L_{tail}$ is a local coherence window that preserves the most recent tokens of the block, which are especially important for maintaining immediate autoregressive context for continued decoding. $K_{min}$ is a count-based lower bound, whereas $L_{tail}$ is a position-based local-context constraint. This is why Eq.3 uses $k_i=\max(K_{\min},L_{\text{tail}},\lfloor r_in_i\rfloor)$, so that the retained-token count satisfies both the minimum block-level budget and the required recent-token window.
>
> We agree that this distinction should have been stated more explicitly before Eq.3. In the revision, we will strengthen the explanation in the Preliminaries and make the different roles of $K_{min}$ and $L_{tail}$ explicit before presenting the equation.
>
> >The notation for $|T_i|$, $L_{tail}$, and $k_i$ is confusing and redundant.
>
> **(1)$|T_i|$ and $L_{tail}$**
>
> Our intention is to explicitly define $L_{tail}$ so that its physical meaning is clear: it represents the length of the mandatory tail segment that must be preserved for each block, rather than just an abstract constant in the formula. The notation $|T_i|$ was introduced to make $L_{tail}$ more concrete and intuitive at the set level. We agree that the notation can be further streamlined. In the revised version, we will optimize the notation.
>
> **(2)max function in $m_i$**
>
> The max function was intentionally included for mathematical rigor and algorithmic robustness. We agree that the preceding minimum constraint ($k_i \ge L_{tail}$) guarantees that the remaining budget is non-negative, given that $|T_i| = L_{tail}$. However, from an engineering perspective, this term acts as a strict defensive bound. It ensures that the heavy-hitter token count never evaluates to a negative value during implementation, protecting against extreme edge cases such as custom parameter overrides, floating-point rounding artifacts, or dynamic budget throttling under severe memory pressure. We will add a brief note in the text to clarify this defensive design.
>
> **(3)$k_i$ and $L_{tail}$**
>
> $k_i$ denotes the total number of tokens retained for block $i$. By contrast, $L_{tail}$ does not refer to previously retained tokens; it refers to the last consecutive token positions of the block itself. That is, among the $k_i$ retained tokens, the final $L_{tail}$ tokens in the original block span must always be preserved. $k_i$ is the overall retention budget, while $L_{tail}$ is a positional constraint that guarantees preservation of the block’s most recent local context.
>
> >Too many notations and terms are used.
>
> Thank you for this comment! We introduce these notations to explicitly represent different components of the problem. We will further simplify the exposition and add a notation table in the appendix to summarize the meaning of key variables this to improve readability.
>
> **We sincerely thank the reviewer again and hope these clarifications help address your concerns! If so, we would deeply appreciate it if you could consider raising your score.** Regardless, we truly appreciate your time and constructive feedback.

---

> > ### Author Rebuttal · Reviewer_s5J8 · 2026-04-04
> >
> > The clarifications in the rebuttal are helpful. But since the readers will not have the same chance as the reviewer to get these clarifications, I strongly suggest the authors consider modifying the technical details with the comments of the reviewer.

---

> > > ### Author Response · Authors · 2026-04-04
> > >
> > > Dear Reviewer s5J8,
> > >
> > > Thank you for your kind support! We sincerely appreciate your valuable comments. We will further polish the presentation in the final version based on your suggestions. We believe these suggestions will further strengthen the quality of the paper.
> > >
> > > With gratitude,
> > >
> > > Authors

---

### Decision · Program_Chairs · 2026-04-30

**Decision:**

Accept (regular)

**Comment:**

This paper proposes ArborKV, a structure-aware KV cache eviction framework tailored for Tree-of-Thoughts (ToT) style LLM reasoning. Unlike traditional sequence-centric methods, ArborKV leverages the structural properties of tree-based search by coupling a Multi-Signal Value Estimator (MSVE) to score thought blocks with a Tree-Aware Eviction (TAE) policy based on topological proximity. By incorporating token-extractive eviction and lazy rehydration for backtracking, the proposed method achieves significant KV memory reduction while maintaining near-full-retention reasoning accuracy across multiple mathematical and logical reasoning benchmarks.

The reviewers universally recognized the significance and timeliness of addressing KV cache management specifically for tree-structured LLM reasoning, noting that it tackles a very real and underserved bottleneck in modern inference systems (s5J8, KDGU). The conceptual framing of the problem—shifting from purely sequence-centric allocation to a topology-aware approach—was highly praised as an intuitive, natural, and promising direction (s5J8, KDGU). Furthermore, the clear algorithmic separation of the value estimator from the allocation mechanism, supported by an elegant optimization view with a KKT-derived closed-form solution, provides a strong theoretical grounding that impressed the committee (KDGU, RnsK). The empirical evaluation was also highlighted as comprehensive and convincing, successfully demonstrating substantial memory reduction with minimal performance degradation (s5J8, RnsK).

Despite these strong conceptual contributions, reviewers raised several concerns regarding presentation clarity, engineering feasibility, and baseline fairness, practical deployment challenges and so on. Authors provided comprehensive rebuttal to address them.


Overall, the paper is recommended to be accepted.